# Functional CRISPR screening identifies the ufmylation pathway as a regulator of SQSTM1/p62

Rowena DeJesus[1†], Francesca Moretti[2†], Gregory McAllister[1], Zuncai Wang[1], Phil Bergman[1], Shanming Liu[1], Elizabeth Frias[1], John Alford[1], John S Reece-Hoyes[1], Alicia Lindeman[1], Jennifer Kelliher[1], Carsten Russ[1], Judith Knehr[2], Walter Carbone[2], Martin Beibel[2], Guglielmo Roma[2], Aylwin Ng[3], John A Tallarico[1], Jeffery A Porter[1], Ramnik J Xavier[3], Craig Mickanin[1], Leon O Murphy[1], Gregory R Hoffman[1*], Beat Nyfeler[2*]

[1]Developmental and Molecular Pathways, Novartis Institutes for BioMedical Research, Cambridge, United States; [2]Developmental and Molecular Pathways, Novartis Institutes for BioMedical Research, Basel, Switzerland; [3]Gastrointestinal Unit and Center for the Study of Inflammatory Bowel Disease, Massachusetts General Hospital, Harvard Medical School, Boston, United States

**Abstract** SQSTM1 is an adaptor protein that integrates multiple cellular signaling pathways and whose expression is tightly regulated at the transcriptional and post-translational level. Here, we describe a forward genetic screening paradigm exploiting CRISPR-mediated genome editing coupled to a cell selection step by FACS to identify regulators of SQSTM1. Through systematic comparison of pooled libraries, we show that CRISPR is superior to RNAi in identifying known SQSTM1 modulators. A genome-wide CRISPR screen exposed MTOR signalling and the entire macroautophagy machinery as key regulators of SQSTM1 and identified several novel modulators including HNRNPM, SLC39A14, SRRD, PGK1 and the ufmylation cascade. We show that ufmylation regulates SQSTM1 by eliciting a cell type-specific ER stress response which induces SQSTM1 expression and results in its accumulation in the cytosol. This study validates pooled CRISPR screening as a powerful method to map the repertoire of cellular pathways that regulate the fate of an individual target protein.

*For correspondence: greg. hoffman@novartis.com (GRH); beat.nyfeler@novartis.com (BN)

†These authors contributed equally to this work

## Introduction

SQSTM1 is a multifunctional adaptor protein that is mutated and deregulated in several pathological conditions, including Paget disease, neurodegeneration, metabolic dysfunction or cancer (*Laurin et al., 2002*; *Rodriguez et al., 2006*; *Rubino et al., 2012*; *Takamura et al., 2011*). As a scaffolding protein, SQSTM1 regulates different cellular signaling cascades such as nuclear factor-κB (NF-κB) activation or the mammalian target of rapamycin complex 1 (mTORC1) (*Duran et al., 2011*; *Katsuragi et al., 2015*). Furthermore, SQSTM1 plays a critical role in autophagy, a catabolic process that results in the lysosomal degradation of cytosolic components (*He and Klionsky, 2009*; *Mizushima and Komatsu, 2011*). In autophagy, SQSTM1 functions as an adaptor that recruits ubiquitinated cargo into forming autophagosomes via its LC3 interacting domain (*Bjørkøy et al., 2005*; *Stolz et al., 2014*). Cargo along with SQSTM1 is then trafficked into lysosomes for degradation. Intracellular levels of SQSTM1 are controlled through turnover by autophagy but also at the transcriptional level upon insults such as oxidative, proteotoxic or endoplasmic reticulum (ER) stress (*Jain et al., 2010*; *Kageyama et al., 2014*; *Liu et al., 2012*).

Here we describe a forward genetic screening approach for novel regulators of SQSTM1 exploiting CRISPR-Cas9 for genome editing. The CRISPR-Cas9 system relies on single guide RNAs (sgRNAs) to target the endonuclease enzyme Cas9 to a genetic locus where it cleaves DNA (*Cong et al., 2013*; *Sternberg et al., 2014*). Repairing of the double strand break by non-homologous end joining results in the insertions and deletions of base pairs often causing loss-of-function phenotypes when targeted to coding regions. We first compared CRISPR to a previously established pooled RNAi approach (*Eng et al., 2016*; *Hoffman et al., 2014*) by screening focused libraries of sgRNA and shRNA reagents and found that CRISPR is superior in identifying known modulators of SQSTM1. In light of this validation data, we carried out a genome-wide CRISPR screen, which robustly uncovered the core macroautophagy machinery and identified several novel regulators of SQSTM1. One example is ubiquitin fold modifier 1 (UFM1) and its conjugation machinery which we validated as *bona fide* modulators of SQSTM1. The mechanism of SQSTM1 accumulation upon impaired ufmylation is autophagy-independent and relies on the activation of the unfolded protein response (UPR) inducing SQSTM1 expression and its accumulation in the cytosol. This study shows that FACS-based pooled CRISPR screening is a powerful forward genetic tool for interrogating complex signaling networks and can be applied for the discovery of cellular targets that modulate protein fate which holds great promise for disease-causing candidates such as oncogenes in cancer or misfolded proteins linked to proteinopathies.

## Results and discussion

### Optimization of a FACS-based pooled CRISPR screen for modulators of SQSTM1

To enable a pooled CRISPR screen for modulators of SQSTM1, we established a neuroglioma H4 cell line which stably expresses Cas9 and green fluorescent protein (GFP)-tagged SQSTM1 (H4 Cas9 GFP-SQSTM1 cells). The cell line was validated by analyzing GFP-SQSTM1 upon autophagy pathway perturbation with the CRISPR-Cas9 system. Induction of autophagy with sgRNAs targeting MTOR decreased GFP-SQSTM1 fluorescence, while inhibition of autophagy with sgRNAs against the lipid kinase PIK3C3 (also known as VPS34) robustly accumulated GFP-SQSTM1 (*Figure 1A*). Using this cell line, we developed a pooled CRISPR screening workflow which relies on isolating cell populations based on their GFP fluorescence by FACS (*Figure 1B*). In brief, pooled sgRNA-encoding libraries are introduced into H4 Cas9 GFP-SQSTM1 cells by lentiviral infection, selected for stable integration, and FACS is used to separate cells into GFP high (upper quartile of GFP fluorescence) or GFP low (lower quartile of GFP fluorescence) populations (*Figure 1—figure supplement 1*). Genomic DNA is isolated from these cell populations and analyzed by deep sequencing to determine the abundance of the sgRNA-encoding sequences. To validate the screening workflow, we analyzed the distribution of 4–5 sgRNAs targeting MTOR, PIK3C3, PLK1 and SQSTM1 in GFP high versus GFP low cells at day 7 post-infection (*Figure 1C*). Cytotoxicity was monitored by the representation of sgRNA sequences in unsorted cells versus the input library. Consistent with autophagy modulation, MTOR sgRNAs were enriched in GFP low cells while PIK3C3 sgRNAs were enriched in GFP high cells. MTOR and PIK3C3 sgRNAs showed an unchanged distribution in unsorted cells versus the input library suggesting that these reagents did not significantly impair cell proliferation. We also expected SQSTM1 sgRNAs to be enriched in GFP low cells due to a knockout of the GFP-SQSTM1 reporter. However, we observed an equal distribution in GFP high and GFP low cells and found that SQSTM1 sgRNA sequences were under-represented in unsorted cells versus the input library, similar to sgRNAs against the essential gene PLK1. Cas9-mediated cytotoxicity has been reported when targeted to regions of gene amplification (*Wang et al., 2015*; *Munoz et al., 2016*). Therefore it is likely that SQSTM1 sgRNAs target both endogenous SQSTM1 and multiple GFP-SQSTM1 chromosome integrants in the H4 reporter clone which results in cytotoxicity in response to DNA damage. We conclude that our pooled CRISPR screening strategy in H4 Cas9 GFP-SQSTM1 cells can identify modulators of SQSTM1 and discriminate cytotoxic hits.

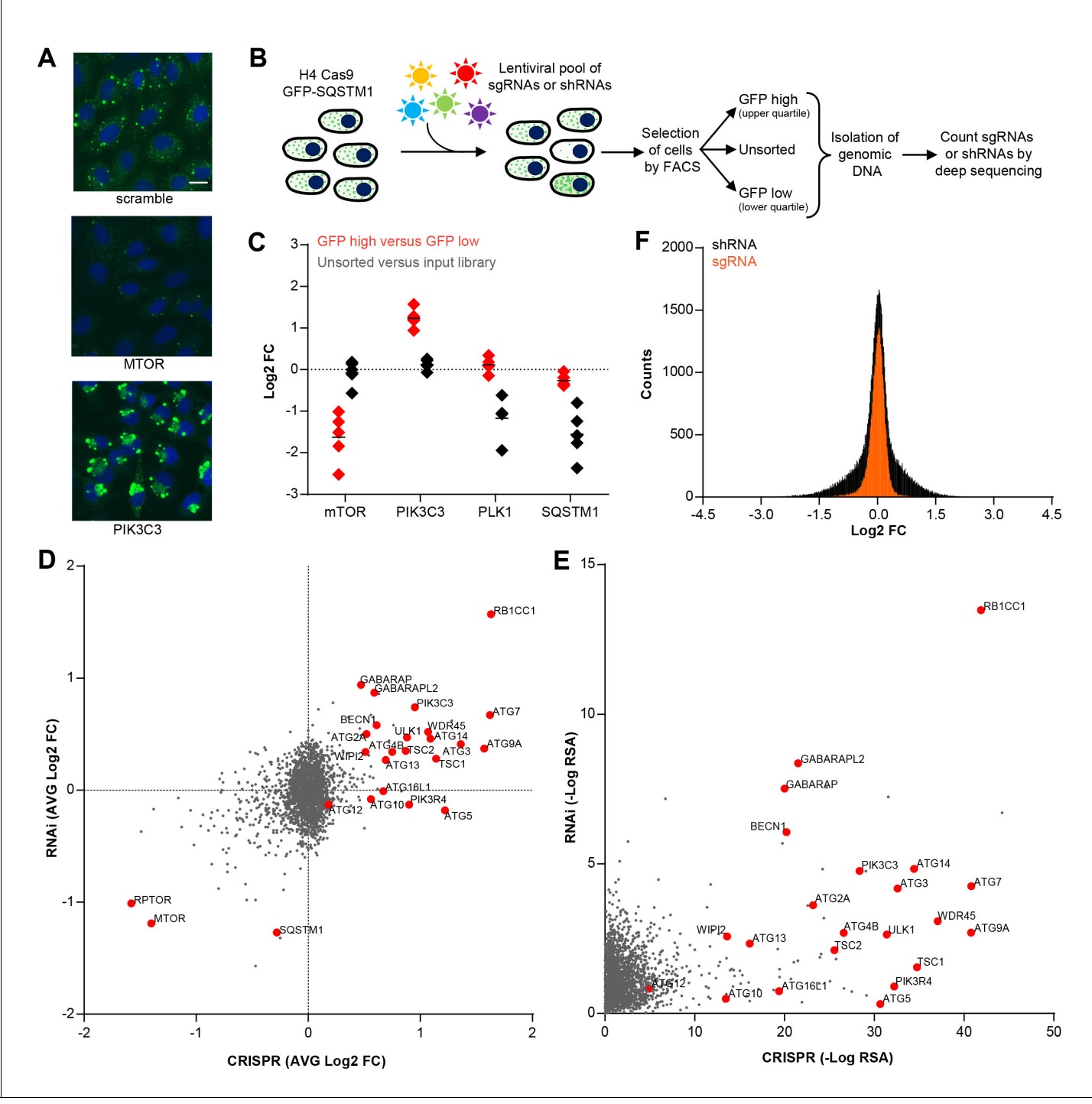

**Figure 1.** CRISPR-based screening outperforms RNAi in identifying modulators of SQSTM1. (**A**) Scramble, MTOR and PIK3C3 sgRNAs were introduced into H4 Cas9 GFP-SQSTM1 cells by lentiviral infection and GFP fluorescence was analyzed after 7 days. Representative images show nuclei (blue) and GFP-SQSTM1 (green). Scale bar corresponds to 20 μm. (**B**) Pooled screening workflow of GFP-SQSTM1 assay. H4 Cas9 GFP-SQSTM1 cells are transduced with lentiviral libraries of sgRNAs or shRNAs and selected for stable integration. Fluorescence activated cell sorting (FACS) is used to isolate cell populations based on GFP upper quartile fluorescence (GFP high) or GFP lower quartile fluorescence (GFP low). A representative GFP FACS histogram is shown in *Figure 1—figure supplement 1*. Abundance of sgRNA and shRNA sequences is quantified by deep sequencing of the corresponding barcodes in the genomic DNA of the isolated cell populations as well as unsorted cells. (**C**) Distribution of individual sgRNAs targeting MTOR, PIK3C3, PLK1 or SQSTM1 at day 7. GFP-SQSTM1 modulation was assessed as log2 fold ratio of each sgRNA sequence based on the abundance in the GFP high versus GFP low cell population. Anti-proliferative effects were assessed as log2 fold ratio of each sgRNA based on the abundance in unsorted cells versus the input library. (**D–F**) H4 Cas9 GFP-SQSTM1 cells were transduced with CRISPR or RNAi lentiviral libraries covering

*Figure 1 continued on next page*

*Figure 1 continued*

2677 genes with an average of 20 sgRNA or shRNA reagents per gene. Both screens were run in duplicate and the mean is shown. Gene-centric visualization of (D) average log2 fold change (FC) or (E) redundant siRNA activity (RSA) scores in GFP high versus GFP low cells. Selected autophagy and MTOR pathway components are highlighted in red. (F) Log2 fold change (FC) distribution of all shRNAs and sgRNAs in GFP high versus GFP low cells.

The following figure supplement is available for figure 1:

**Figure supplement 1.** GFP FACS histogram highlighting the gates used to sort GFP high and GFP low cell populations.

## Pooled CRISPR screening is superior to RNAi in identifying known regulators of SQSTM1

To compare CRISPR to a previously established pooled RNAi approach (*Eng et al., 2016*; *Hoffman et al., 2014*), we screened focused deep-coverage libraries of sgRNA and shRNA reagents in H4 Cas9 GFP-SQSTM1 cells using the FACS-based workflow outlined in *Figure 1B*. Each library pool covered 2677 genes, including many autophagy core components, with an average of 20 sgRNAs and 20 shRNAs per gene. When the average fold change per gene was analyzed, both CRISPR and RNAi identified many known modulators of SQSTM1, including MTOR, RPTOR, RB1CC1 (also known as FIP200), BECN1, PIK3C3, ULK1 or ATG7 (*Figure 1D*). However, several well-characterized modulators of SQSTM1 including ATG5, ATG10, ATG13 or ATG16L1 were exclusively identified by CRISPR. A complete knockout of these targets by CRISPR may be needed to robustly impair the autophagic clearance of SQSTM1, in line with previous reports (*Hosokawa et al., 2006*; *Mizushima et al., 2010*) suggesting that some autophagy proteins, including ATG5, are functional even when expressed at very low levels. Moreover, CRISPR outperformed RNAi in a significance analysis using redundant siRNA activity (RSA) statistics (*König et al., 2007*) where more known modulators of SQSTM1 scored with statistically significant p-values by CRISPR (*Figure 1E*). In comparison to shRNAs, sgRNAs showed a tighter distribution when all CRISPR and RNAi reagents were analyzed in GFP high versus GFP low cells (*Figure 1F*). Taken together, this data suggests that CRISPR is superior to RNAi in separating hits from the bulk of inactive reagents in these libraries.

## A genome-wide screen for modulators of SQSTM1 robustly identifies canonical macroautophagy components

To identify modulators of SQSTM1 at genome-wide scale, we screened a sgRNA library covering 18,360 genes with an average of 5 sgRNAs per gene in the FACS-based GFP-SQSTM1 assay (*Figure 2A–E*). By analyzing sgRNA sequences enriched in GFP low cells, we identified MTOR, RPTOR and RHEB among the top ranked genes, in line with their depletion activating autophagy and inducing degradation of SQSTM1 (*Figure 2A*). We also observed several components of the mediator complex and an enrichment analysis of the top ranked 100 genes revealed transcriptional regulation by RNA polymerase II as significantly enriched GO biological process (*Figure 2C*). CRISPR-mediated depletion of these mediator complex components likely decreases GFP-SQSTM1 by reducing reporter expression, a hypothesis which we evaluated with the mini-pool follow-up approach discussed below.

sgRNA sequences enriched in GFP high cells targeted many autophagy genes such as ATG9A, RB1CC1, ATG16L1, ATG7, ATG3, ATG5, WIPI2, MAP1LC3B and ATG14 (*Figure 2B*). Autophagy-related categories were the most significantly enriched GO biological processes when the top ranked 100 genes in GFP high cells were analyzed (*Figure 2D*). The genome-wide GFP-SQSTM1 CRISPR screen comprehensively mapped the canonical macroautophagy pathway and identified components of the initiation, nucleation and elongation machinery (*Figure 2E*). In comparison to the 20 sgRNAs per gene in the focused CRISPR library (*Figure 1*), the 5 sgRNA per gene in the genome-wide library performed equally well in identifying autophagy pathway components when the maximal fold change was analyzed per gene (*Figure 2—figure supplement 1*). Autophagy pathway components which did not significantly score were restricted to genes which have functional homologs. These include ATG4A, ATG4C, ATG4D, ULK2, ATG16L2, WIPI1 and ATG9B which were all inactive while their homologs ATG4B, ULK1, ATG16L1, WIPI2 and ATG9A robustly scored. By

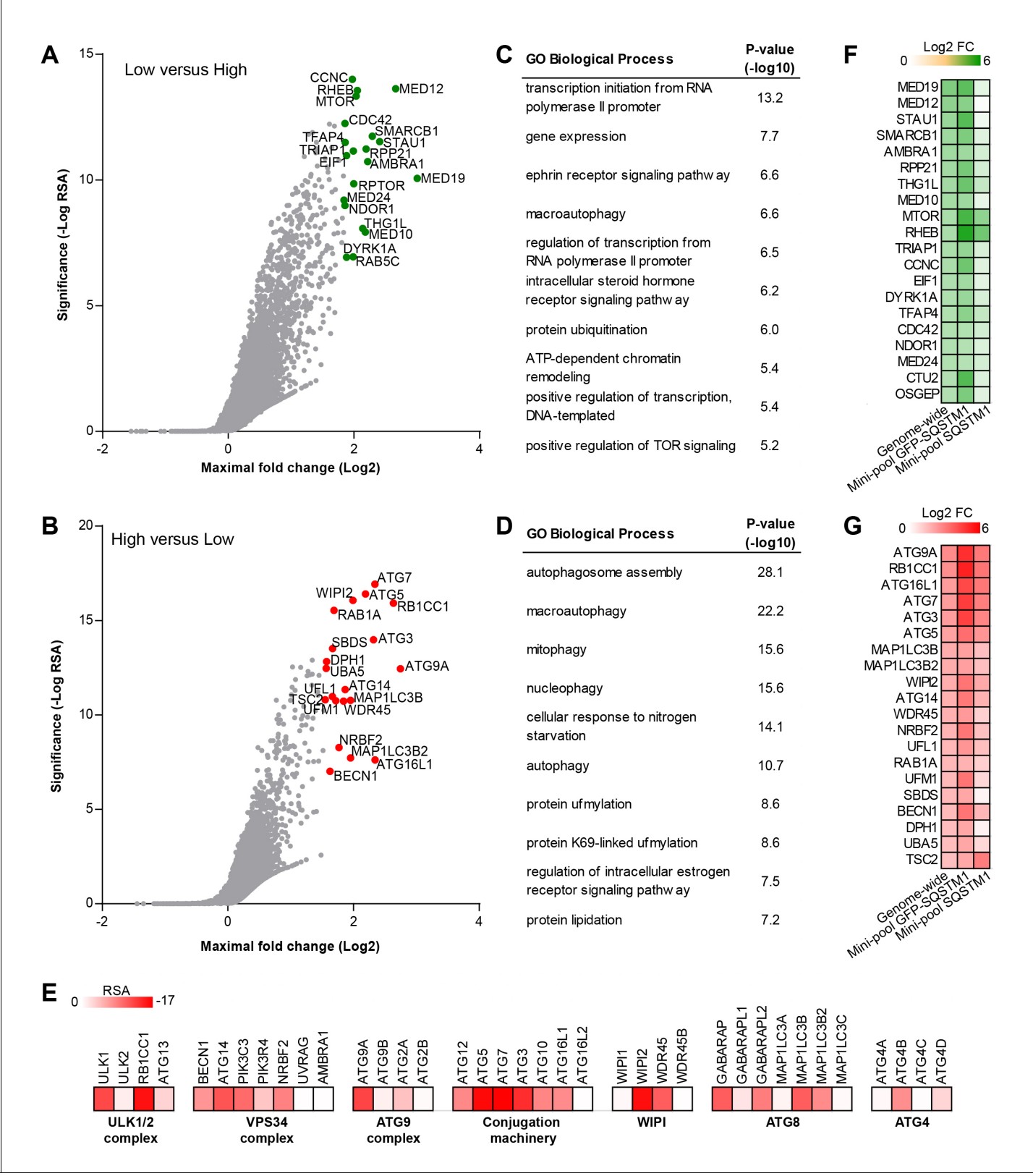

**Figure 2.** Genome-wide pooled CRISPR screening dissects canonical macroautophagy. (A–E) H4 Cas9 GFP-SQSTM1 cells were transduced with a genome-wide CRISPR library covering 18,360 genes with an average of 5 sgRNAs per gene. 7 days post-infection, FACS was used to isolate GFP high and GFP low cell populations, and abundance of sgRNA sequences was quantified by deep sequencing. Gene-centric visualization of significance and
*Figure 2 continued on next page*

*Figure 2 continued*

maximal fold change in (A) GFP low versus GFP high cells, or (B) GFP high versus GFP low cells. Top scoring genes are highlighted and entire dataset is reported in *Figure 2—source data 1*. Gene enrichment analysis of the top ranked 100 hits in (C) GFP low versus GFP high cells, or (D) GFP high versus GFP low cells. (E) Heatmap of the RSA significance scores in GFP high versus GFP low cells for core components of canonical macroautophagy. (F–G) A mini-pool of sgRNAs was picked from the genome-wide screen targeting the top ranked 600 hits in GFP high and GFP low cells. The mini-pool was screened in H4 Cas9 GFP-SQSTM1 cells based on GFP fluorescence and in the parental H4 Cas9 cell population based on endogenous SQSTM1 staining. Heatmap of log2 fold changes (FC) for top ranked hits in (F) low versus high cells, or (G) high versus low cells. Hits are ranked based on log2 FC in the genome-wide screen. Entire mini-pool data is reported in *Figure 2—source data 2*.

The following source data and figure supplements are available for figure 2:

**Source data 1.** RSA and maximal log2 fold changes for genome-wide GFP-SQSTM1 screen.

**Source data 2.** Mini-pool sgRNA sequences and corresponding log2 fold changes in genome-wide GFP-SQSTM1, mini-pool GFP-SQSTM1 and mini-pool endogenous SQSTM1 screens.

**Figure supplement 1.** Comparison of the focused versus genome-wide sgRNA libraries.

**Figure supplement 2.** High-level concept map of enriched pathways and processes for screen hits.

analyzing subclusters of highly connected gene sets, a strong enrichment and connectivity signal was confirmed for several pathways and processes, including autophagy and MTOR signaling (*Figure 2— figure supplement 2*).

To validate the results from the primary screen, we generated a mini-pool containing the single most potent sgRNA-encoding sequences from the primary screen against the top-ranked 300 genes in the GFP low and GFP high category. Screening of the mini-pool in H4 Cas9 GFP-SQSTM1 cells showed a high confirmation rate when the top-ranked hits from the primary screen were analyzed (*Figure 2F and G*). To exclude GFP reporter-specific hits, we used the mini-pool to screen a parental H4 Cas9 cell population followed by immunofluorescence-based staining of endogenous SQSTM1. Similar to the GFP assay, we used FACS to isolate a high and low cell population based on the SQSTM1 staining intensity. This approach confirmed that the MTOR and autophagy pathways are key regulators of SQSTM1. However, components of the mediator complex or targets linked to translation such as SBDS and DPH1 failed to robustly score which suggests that these specifically regulate the GFP-SQSTM1 reporter (*Figure 2F and G*).

## Identification of novel regulators of SQSTM1

We analyzed the mini-pool data for novel modulators of SQSTM1. sgRNAs against HNRNPM, SLC39A14, SRRD and PGK1 were enriched in cells displaying high levels of both GFP-SQSTM1 and endogenous SQSTM1 (*Figure 3A*), and to our knowledge, these candidates have not been linked to the regulation of SQSTM1. When these sgRNAs were individually tested in H4 Cas9 cells, we observed a robust increase in SQSTM1 levels by immunoblot, similar to the depletion of ATG7 (*Figure 3B*). This data demonstrates that the FACS-based pooled CRISPR results can be confirmed at the level of individual sgRNAs.

We also noticed that ubiquitin-fold modifier 1 (UFM1) and components of its conjugation machinery scored as modulators of endogenous and GFP-tagged SQSTM1 (*Figure 3A*). UFM1 is a ubiquitin-like molecule which is post-translationally processed by UFM1-specific proteases (UFSP1 and UFSP2) and then conjugated to target proteins via the ubiquitin-like modifier activating enzyme 5 (UBA5), ubiquitin-fold modifier conjugating enzyme 1 (UFC1) and the UFM1-specific ligase (UFL1) (*Figure 3C*) (*Daniel and Liebau, 2014*; *Komatsu et al., 2004*). Protein ufmylation was already exposed in the primary screen as one of the most significantly enriched biological processes in GFP-SQSTM1 high cells (*Figure 2D*) and all components of the cascade scored with the exception of UFSP1. Ufmylation has been implicated in the control of cell differentiation, growth and endoplasmic reticulum (ER) homeostasis (*Daniel and Liebau, 2014*). Knockout of the murine homolog of UFL1 has been linked to impaired autophagy (*Zhang et al., 2015*), but the role of ufmylation in the autophagic process is controversial (*Cai et al., 2015*). To confirm ufmylation as modulator of SQSTM1,

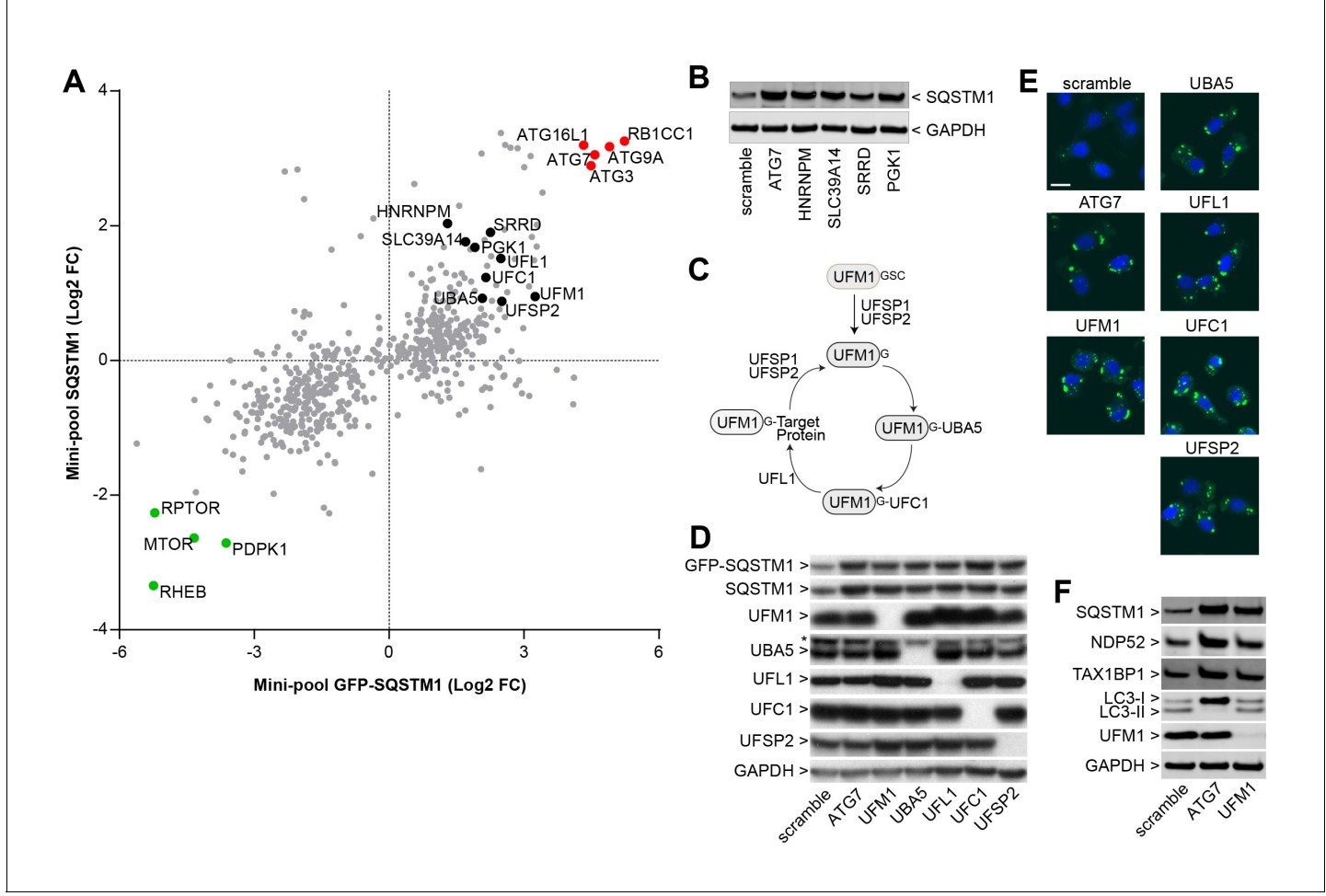

**Figure 3.** Identification of novel modulators of SQSTM1. (**A**) Comparison of log2 fold changes in the mini-pool screen for GFP-SQSTM1 and endogenous SQSTM1. Selected MTOR and autophagy pathway genes are marked in green or red while novel candiates are highlighted in black. (**B**) Indicated sgRNAs were introduced into H4 Cas9 cells by lentiviral infection, protein lysates were collected 7 days post-infection and probed by immunoblot. (**C**) Cartoon of the ufmylation cascade. (**D–E**) Accumulation of GFP-SQSTM1 upon depletion of ufmylation. Indicated sgRNAs were introduced into H4 Cas9 GFP-SQSTM1 cells by lentiviral infection and cells were probed 7 days post-infection by (**D**) immunoblot or (**E**) high-content imaging. Representative images show nuclei (blue) and GFP-SQSTM1 (green). Scale bar corresponds to 20 μm. (**F**) Indicated sgRNAs were introduced into H4 Cas9 clone 4 by lentiviral infection, protein lysates collected 7 days post-infection and probed by immunoblot.

we individually tested sgRNAs against UFM1, UBA5, UFL1, UFC1 and UFSP2 and confirmed efficient depletion of the corresponding target proteins (*Figure 3D*) as well as robust accumulation of GFP-SQSTM1 and endogenous SQSTM1 (*Figure 3D–F*).

## Impairment of protein ufmylation elicits an ER stress response and induces SQSTM1 expression

We next sought to elucidate the mechanism by which ufmylation regulates SQSTM1. Depletion of UFM1 increased SQSTM1 and slightly augmented the levels of the autophagy cargo receptors CAL-COCO2 (also known as NDP52) and TAX1BP1; however, levels of LC3-I and LC3-II appeared largely unchanged (*Figure 3F*). UFM1-depleted cells were as sensitive to the V-ATPase inhibitor Bafilomycin A1 as the control cells suggesting that autophagic flux of LC3 is unchanged (*Figure 4A*). In contrast to LC3B, we did not see lysosomal localization of UFM1 when tagged with the mCherry-GFP tandem reporter (*Nyfeler et al., 2012*) (*Figure 4B*) suggesting that UFM1 is not an alternative lysosomal tar-geting signal. Moreover, while ATG7-depleted cells were unable to degrade GFP-SQSTM1, UFM1 knockout cells still cleared GFP-SQSTM1 when autophagy was activated by the MTOR inhibitor

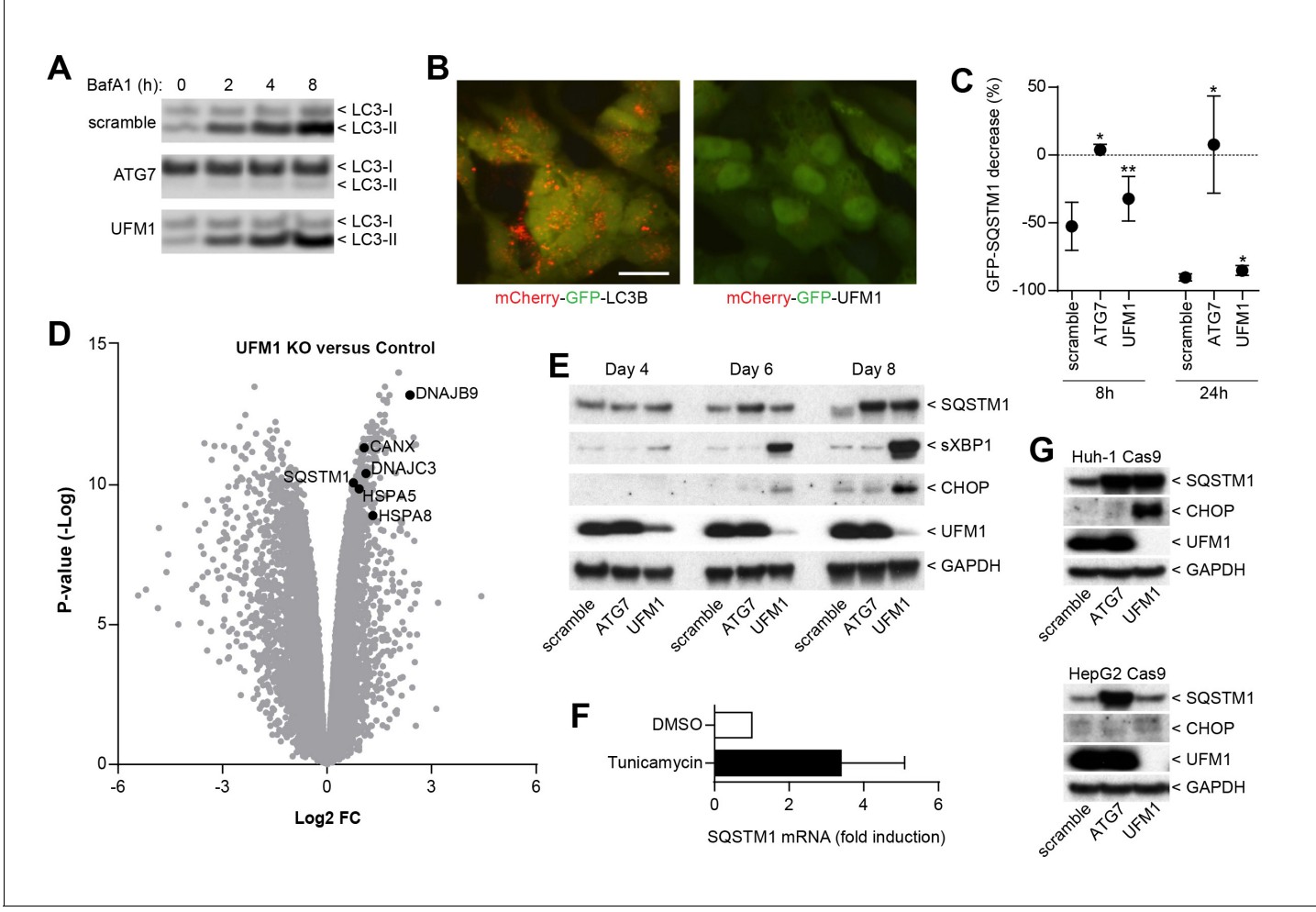

**Figure 4.** Inhibition of ufmylation elicits an ER stress response and induces SQTSM1 expression. (**A**) Scramble, ATG7 and UFM1 sgRNAs were introduced into H4 Cas9 clone 4 by lentiviral infection. 7 days post-infection cells were treated with 0.1% DMSO or 50 nM Bafilomycin A1 for 2, 4 or 8 hr and protein lysates were collected and probed by immunoblot. (**B**) Representative images of H4 cells stably expressing mCherry-GFP-LC3B or mCherry-GFP-UFM1. Scale bar corresponds to 20 µm. Note that the absence of mCherry-positive puncta in mCherry-GFP-UFM1 expressing cells suggests that UFM1 is not delivered to endolysosomal compartments. (**C**) GFP-SQSTM1 clearance in ATG7- and UFM1-depleted cells. Scramble, ATG7 and UFM1 sgRNAs were introduced into H4 Cas9 GFP-SQSTM1 cells by lentiviral infection. 7 days post-infection cells were treated with 0.1% DMSO or 0.5 µM AZD8055 for 8 hr or 24 hr, fixed, and subjected to high content imaging. GFP fluorescence was quantified as ratio of GFP-SQSTM1 puncta area in DMSO- versus AZD8055-treated cells and is shown as mean +/- SD from three independent experiments. Statistical significance was assessed via a two-tailed paired t-test (*p<0.05, **p<0.01). (**D**) UFM1 knockout induces ER stress and increases SQSTM1 mRNA levels. Scramble and UFM1 sgRNAs were introduced into H4 Cas9 cells by lentiviral infection. 7 days post-infection total RNA was extracted and subjected to RNA sequencing. The volcano plot visualizes genes differentially expressed in UFM1-depleted versus cells infected with a scramble sgRNA. Selected ER stress markers and SQSTM1 are highlighted and entire dataset is reported in *Figure 4—source data 1*. (**E**) Time course analysis of ER stress and SQSTM1 induction. Indicated sgRNAs were introduced into H4 Cas9 cells by lentiviral infection, protein lysates collected 4, 6 and 8 days post-infection and probed by immunoblot. (**F**) Total RNA was collected from H4 Cas9 cells treated with 0.1% DMSO or 10 µg/ml tunicamycin for 24 hr, SQSTM1 mRNA was quantified by RT-qPCR analysis and normalized to HPRT1. The mean +/- SD of biological triplicates is shown. (**G**) Indicated sgRNAs were introduced into Huh-1 Cas9 or HepG2 Cas9 cells by lentiviral infection, protein lysates collected 7 days post-infection and probed by immunoblot.

The following source data is available for figure 4:

**Source data 1.** Fold change and significance of differentially regulated genes by RNA sequencing in UFM1-depleted versus scramble sgRNA-infected cells.

AZD8055, albeit with a reduced rate than cells infected with a scramble sgRNA control (*Figure 4C*). Taken together, we conclude that autophagy-independent mechanisms likely contribute to the increase in SQSTM1 in UFM1-depleted cells.

Since ufmylation has been implicated in the control of transcription factor activity (*Yoo et al., 2014*), we analyzed the RNA expression profile upon UFM1 depletion in H4 cells. Compared to control sgRNA-infected cells, 526 genes were down-regulated while 276 genes were up-regulated more than twofold (p-value $<10^{-5}$) when UFM1 was deleted. A gene set enrichment analysis identified the ER unfolded protein response as most significantly enriched biological process for the upregulated genes. Similar to known ER stress markers such as HSPA5 (also known as BiP) or CANX, we observed that knockout of UFM1 increased SQSTM1 expression (~ 1.7-fold, *Figure 4D*). Induction of ER stress by tunicamycin also increased mRNA levels of SQSTM1 (*Figure 4F*), in line with SQSTM1 being a target of the UPR (*Liu et al., 2012*). In a UFM1-depletion time course experiment, accumulation of SQSTM1 was preceded by the induction of ER stress as measured by XBP1 splicing and CHOP (*Figure 4E*). Interestingly, UFM1 knockout induced CHOP and SQSTM1 also in Huh-1 but not in HepG2 cells which did not show any signs of ER stress or SQSTM1 accumulation despite efficient depletion of UFM1. This data suggests that impairment of ufmylation results in a cell type-specific ER stress response which in turn induces SQSTM1 expression and its accumulation in the cytosol. SQSTM1 has been shown to act as a key regulator of the cytosolic response to various cellular insults by forming aggresome-like induced structures (ALIS) which incorporate ubiquitinated proteins and heat shock proteins (*Liu et al., 2012*). We hypothesize that other ubiquitin-binding cargo receptors such as NDP52 or TAX1BP1 may also get recruited into ALIS thereby accumulating in UFM1 knockout cells (*Figure 3F*). Alternatively, the increase in SQSTM1 in UFM1-depleted cells may induce competition for the autophagy machinery resulting in other autophagy cargo receptors being less efficiently cleared. Given the interaction of UBA5 and ATG8 homologs (*Habisov et al., 2016*), ufmylation may directly link autophagy to the regulation of the ER stress response and future studies will be needed to test if transcription factors of the unfolded protein response are directly modified by UFM1.

In summary, this study demonstrates that CRISPR-Cas9 genome editing can be exploited as a robust forward genetic screening approach to identify *bona fide* modulators of protein fate when combined to a phenotypic selection step by FACS. The systematic comparison of RNAi and CRISPR with focused libraries revealed a superior discovery rate of known SQSTM1 modulators by CRISPR (*Figure 1D and 1E*). Consistent with our observation, CRISPR has been reported to outperform RNAi in identifying essential genes (*Munoz et al., 2016*; *Evers et al., 2016*). Like RNAi, CRISPR has intrinsic limitations such as potential off-target editing (*Fu et al., 2013*; *Hsu et al., 2013*), and FACS-based phenotypic screens such as GFP-SQSTM1 will be restricted to probe for non-essential modulators. Hence, RNAi and CRISPR-based approaches have to be evaluated depending on the screening paradigm, and combined parallel approaches may help to mitigate the intrinsic limitations of both technologies (*Deans et al., 2016*; *Morgens et al., 2016*). By following-up the genome-wide GFP-SQSTM1 CRISPR screen with a mini-pool screen for modulators of endogenous SQSTM1, we exposed MTOR complex 1 and canonical macroautophagy as key regulators of SQSTM1 (*Figure 3A*). Furthermore, this study identified novel modulators of SQSTM1 (*Figure 3B*) including the entire biochemical cascade of the ufmylation process (*Figure 3A and C*), thus providing a resource to further unravel the regulation of SQSTM1.

## Materials and methods

### Mammalian cell culture

H4 (ATCC, HTB-148), HepG2 (ATCC, HB-8065) and Huh-1 (JCRB, JCRB0199) cell lines were confirmed by SNP analysis and verified to be mycoplasma-free by routine testing. All cells were kept in a humidified incubator at 37°C and 5% $CO_2$ and were maintained in Dulbecco's modified Eagle's medium (DMEM) supplemented with 10% FBS, 1% L-glutamine and 1% Penicillin/Streptomycin. Cell culture reagents were obtained from Invitrogen. H4 GFP-SQSTM1 cells were generated by retroviral delivery of EGFP-SQSTM1 (*Dowdle et al., 2014*) in pRetro-X Tight (Clontech, Mountain View, CA) into human neuroglioma H4 cells followed by the isolation of GFP-positive cells by FACS. pRetro-X Tight was modified to eliminate puromycin resistance by removing the EcoRI - RsrII fragment. H4

Cas9 GFP-SQSTM1 cells were generated by lentiviral delivery of Cas9 in pNGx-LV-c004 (*Supplementary file 1A*) into H4 GFP-SQSTM1 cells followed by selection with blasticidin. Individual clones were expanded, tested for Cas9 and GFP-SQSTM1 expression by western blot and immuno-fluorescence, and H4 Cas9 GFP-SQSTM1 clone 2 was picked for the screen and validation experiments. H4 Cas9 cells were generated by lentiviral delivery of Cas9 in pNGx-LV-c004 into H4 cells followed by selection with blasticidin. The H4 Cas9 polyclonal population or the isolated clone 4 was used as indicated in the figure legends. HepG2 and Huh1 Cas9 cells were generated by lentiviral delivery of Cas9 in pNGx-LV-c004 followed by selection with blasticidin and isolation of single clones. H4 mCherry-GFP-LC3 and H4 mCherry-GFP-UFM1 lines were generated by lentiviral delivery of mCherry-GFP-LC3 and mCherry-GFP-UFM1 in pLenti6.3 followed by selection for stable integration with blasticidin. Coding sequences for MAP1LC3B and UFM1 were derived from Invitrogen entry clones (IOH13267, IOH7279).

## Pooled RNAi and CRISPR screening

### shRNA and sgRNA library design and construction

For the focused libraries, custom sgRNA and shRNA libraries were constructed by Cellecta as previously described (*Hoffman et al., 2014*). These libraries cover around 2700 genes with an average of 20 unique shRNAs or sgRNAs per gene (*Munoz et al., 2016*). The genome-wide sgRNA library targeting 18,360 protein-coding genes was constructed using chip-based oligonucleotide synthesis to generate spacer-tracrRNA-encoding fragments that were PCR-amplified and cloned as a pool into the BpiI site of the pRSI16 lentiviral plasmid (Cellecta, Mountain View, CA). Olfactory receptors were omitted from the library. The sgRNA designs were based on published sequences (*Wang et al., 2014*) and five sgRNAs were selected per gene targeting the most proximal 5' exons. 277 genes did not have published sgRNA sequence information and new sgRNAs were designed for these targets that contained an NGG PAM motif, filtering for GC content greater than 40% and less than 80%, eliminating homopolymer stretches greater then 4, and removing any guides with off-target locations having fewer than 4 mismatches across the genome. Sequencing of the plasmid pool showed robust normalization with >90% clones present at a representation of +/− five fold from the median counts in the pool.

For the mini-pool library (*Figure 2—source data 2*), sgRNA sequences were picked for the top-ranked 300 genes in either the GFP low or GFP high category from the genome-wide screen. The mini-pool library was constructed using chip-based oligonucleotide synthesis (Custom Array) to generate spacer-encoding fragments that were PCR-amplified and cloned as a pool into the BbsI site of pNGx-LV-g003 lentiviral plasmid (*Supplementary file 1A*). Sequencing of the plasmid pool showed robust normalization with >90% clones present at a representation of +/− fivefold from the median counts in the pool.

### Viral packaging

shRNA and sgRNA libraries were packaged into lentiviral particles using HEK293T cells as described previously (*Hoffman et al., 2014*). Packaging was scaled up by growing cells in cell stacks (Corning, Corning, NY). For each cell stack, 210 million cells were transfected 24 hr after plating using 510.3 µL of TransIT reagent (Mirus, Madison, WI) diluted in 18.4 mL of OPTI-MEM that was combined with 75.6 µg of the shRNA or sgRNA libraries and 94.5 µg of lentiviral packaging mix (Cellecta, psPAX2 and pMD2 plasmids that encode Gag/Pol and VSV-G, respectively). 72 hr post transfection, lentivirus was harvested, aliquoted, and frozen at -80°C. Viral titer was measured by FACS in HCT116 cells and was typically in the range of 5 x $10^6$ TU/mL.

### FACS-based screening procedure

All screens were run in duplicate. shRNA and sgRNA libraries were transduced at an MOI of 0.5 aiming for coverage of on average 1000 cells per shRNA or gRNA reagent. MOI was determined by using a 12-point dose response ranging from 0 to 400 µL of viral supernatants in the presence of 5 µg/mL polybrene and measuring infection rate by FACS as percentage of RFP-positive cells. Selection was optimized by determining the puromycin dose required to achieve >95% cell killing in 72 hr. Cell viability was measured with a Cell Titer Glo assay (Promega, Madison, WI) for a 6-point dose response ranging from 0 to 5 µg puromycin. For the genome-wide screen, 67 million cells were

seeded at 33.5 million cells per cell stack (Corning). 24 hr after plating, the culture media was replaced with fresh media containing 5 µg/mL polybrene and lentivirus at an MOI of 0.5. 24 hr after infection, the culture media was replaced with fresh media containing puromycin. 72 hr after puromycin addition, cells were trypsinized and plated into new cell stacks at 67 million cells per cell stack. An aliquot of cells was analyzed by FACS to confirm infection and selection efficiency, and the percentage of RFP-positive cells was typically >95%. Cells were maintained in culture and split as needed to ensure confluence did not exceed 90%. The sgRNA and shRNA libraries were screened 7 days and 5 days post infection, respectively. For FACS, cells were harvested, resuspended at 30 million cells/mL, and live, single, RFP-positive cells were sorted (BD ARIA III) from the lower GFP quartile (GFP low) or from the upper GFP quartile (GFP high). For the genome-wide screen, 700 million cells were harvested and 50 million cells were sorted by FACS in the GFP low and GFP high category. 50 million unsorted cells were also collected as an input sample.

Cells stained for endogenous SQSTM1 were harvested and fixed in 4% paraformaldehyde. The samples were then washed with PBS, permeabilized in 0.1% Triton X-100 for 10 min, blocked using Odyssey block buffer (LI-COR, Lincoln, NE) for 30 min, and incubated with SQSTM1/p62 primary antibody (BD Biosciences, 1:500) in 1:1 PBS/Odyssey block buffer for 1 hr at room temperature. After washing with PBS, cells were incubated with Alexa647-tagged secondary antibody (Molecular Probes, Waltham, MA), incubated in the dark for 30 min at room temperature, washed again with PBS, strained through a 40 µm mesh filter and resuspended at 20 million cells/mL. Single, RFP-positive cells were sorted (BD ARIAIII) from the lower Alexa647 quartile (SQSTM1 low) or upper Alexa647 quartile (SQSTM1 high). For the mini-pool screen, cells were seeded, infected and selected as previously described for the genome-wide screen. 60 million cells were harvested and prepared for FACS sorting. 2 million cells were sorted by FACS in the SQSTM1 low and SQSTM1 high category, and 2 million cells were collected for the unsorted population.

## Illumina library construction and sequencing

Genomic DNA from live cells was isolated using the QIAamp DNA Blood Maxi kit (Qiagen, Germany) and quantified using PicoGreen (Invitrogen, Waltham, MA) following the manufacturer's instructions. Genomic DNA from 4% paraformaldehyde-fixed cells was isolated using phenol chloroform extraction. Illumina sequencing libraries were generated using PCR amplification with primers specific to the genome integrated lentiviral vector backbone sequence. For the focused shRNA and the genome-wide sgRNA libraries, a total of 24 x 4 µg PCR reactions were performed per transduced sample. For the mini-pool sgRNA library, a total of 4 x 0.25 µg PCR reactions were performed per transduced sample. PCR reactions were performed in a volume of a 100 µl containing a final concentration of 0.5 µM of each PCR primer (Integrated DNA Technologies, Coralville, IA), 0.5 mM dNTPs (Clontech) and 1x Titanium Taq and buffer (Clontech). For the shRNA samples an additional third 'index' PCR primer at a final concentration of 0.01 µM was used. PCR primer sequences for all libraries are shown in *Supplementary file 1B*. PCR cycling conditions used were as follows: 1x 98°C for 5 min; 28x 95°C for 15 s, 65°C for 15 s, 72°C for 30 s; 1x 72°C for 5 min. The resulting Illumina libraries were purified using 1.8x SPRI AMPure XL beads (Beckman Coulter, Indianapolis, IN) following the manufacturer's recommendations and qPCR quantified using primers specific to the Illumina sequences using standard methods. Illumina sequencing libraries were pooled and sequenced with a HiSeq 2500 instrument (Illumina, San Diego, CA). sgRNA libraries were sequenced with 1x30 b (sgRNA) and 1x11 b (sample index) reads. shRNA libraries were sequenced with 1x50 b (shRNA barcode) and 1x9 b (sample index) reads. Sequencing was performed following the manufacturer's recommendations, using custom sequencing primers (see *Supplementary file 1B*). The number of reads was adjusted to cover each sgRNA or shRNA with approximately 1000 reads.

## Data analysis

Raw sequencing reads were aligned to the appropriate library using Bowtie (*Langmead et al., 2009*) allowing for no mismatches and counts were generated. Differential fold change estimates were generated using DESeq2 (*Love et al., 2014*) to compare the representation in the 'low' gate to the 'high' gate in the FACS sort. Additionally, to assess effects on proliferation, the fold change of the unsorted cell population compared to the input library was generated. For gene-based hit calling,

Redundant siRNA Activity (RSA) (*König et al., 2007*) and average or maximal fold changes were calculated across all reagents for a given gene. Biological contextualization was assessed by measuring the significance of overlap between screen hits and Gene Ontology Biological Process groups using a hypergeometric test and correcting for multiple hypothesis testing using Benjamini-Hochberg.

## High-level concept Map of enriched pathways and processes

Statistical enrichment of pathways and process-related gene sets was determined for screen hits using the hypergeometric test implemented in the R programming language. The gene sets were compiled from multiple sources including Reactome, NetPath, KEGG, Wikipathways, BioCarta and Gene Ontology. We applied graph theoretic approaches to the construction of high-level concept maps as described previously (*Anderson et al., 2011*; *Orvedahl et al., 2011*; *Tomlins et al., 2007*). Briefly, the enrichment analyses were abstracted as a network graph, with nodes denoting enriched pathways and processes. In the network, node size scales according to the number of hit genes in each pathway or process, while the node color intensity corresponds to the gene set enrichment score, -Log10(P). For graph edges, we computed Jaccard coefficients to determine similarity between pathways and process-related gene sets enriched among the screen hits. Sub-clusters of strongly connected nodes in the network were identified using Tarjan's algorithm (*Tarjan, 1972*).

## Validation of individual sgRNAs

For each sgRNA expression clone (*Supplementary file 1C*), spacer-encoding sense and antisense oligonucleotides with appropriate overhangs were synthesized (IDT), annealed, cloned into the BbsI restriction site of the pNGx-LV-g003 vector (*Supplementary file 1A*), and verified by sequencing. Lentiviral particles were generated in HEK293T cells using the ViraPower Lentiviral Packaging Mix (Invitrogen) following manufacturer's instructions. H4 Cas9 and H4 Cas9 GFP-SQSTM1 cells were seeded at 250,000 cells/well in 6-well or 100,000 cells/well in 12-well and infected with 400 µL or 200 µL of virus supernatant, respectively. Infected cells were selected with 1 µg/mL puromycin and analyzed seven days post-infection. Where indicated, cells were treated with 50 nM bafilomycin A1 (Tocris, United Kindom), 500 nM AZD8055 (ChemieTek, Indianapolis, IN), 10 µg/mL tunicamycin (Sigma-Aldrich, St. Louis, MO) or 0.1% DMSO.

## RT-qPCR analysis

Cells were lysed and processed using the RNeasy Plus Kit (Qiagen) according to the manufacturer's instructions. cDNA synthesis was performed with the High-Capacity cDNA Reverse Transcription Kit (Applied Biosystems, Waltham, MA). qPCR was performed with the TaqMan Gene Expression Master Mix and the following TaqMan probes from Applied Biosystems: HPRT1 (Hs02800695_m1), SQSTM1 (Hs01061917_g1). The qPCR reactions were run on a ViiA7 real-time PCR machine (Applied Biosystems). Relative quantification of SQSTM1 mRNA was performed using HPRT1 mRNA as normalization control and the $2^{-\Delta\Delta Ct}$ method.

## RNA sequencing

RNA sequencing libraries were prepared using the Illumina TruSeq RNA Sample Prep kit v2 and sequenced using the Illumina HiSeq2500 platform. Samples were sequenced to a length of 2 x 76 base-pairs. A total of 491 million 76 bp paired-end reads were mapped to the Homo sapiens genome (HG19) (*Pruitt et al., 2007*) (release 59, May 3, 2013), and a custom junction database by using an in-house gene and exon quantification pipeline based on the aligner Bowtie (*Langmead and Salzberg, 2012*), version 2.0.2. On average, 95% of the total reads were mapped to the genome or the transcripts, and 90.6% of the aligned reads mapped to expressed sequences. The genome and the transcript alignments were used to derive gene counts based on human Ensembl gene IDs (v78, December 2014). Gene counts were divided by the total number of mapped reads for each sample and multiplied by one million to obtain Counts Per Million (CPMs) to account for varying library sizes. Differential expression analysis was performed on the CPMs using a limma/voom workflow with R version 3.1.0 (*Ritchie et al., 2015*). The statistical model consisted of one factor with two levels for the UFM1 status (control or knockout). Genes with counts per million values below 0.5 for at least two out of three replicates in both groups were excluded from the analysis. The following test was performed: UFM1 knockout versus control. Results are reported in terms of

log2 fold changes and negative log10 adjusted P values (Benjamini Hochberg false discovery rate) in *Figure 4—source data 1*. For the enrichment analysis of differentially regulated genes, we used an in-house implementation of overrepresentation analysis using hypergeometric testing and a Benjamini-Hochberg FDR correction.

## Western blotting

Cells were lysed in RIPA buffer (Thermo Scientific, Waltham, MA) supplemented with protease inhibitor cocktail (Roche) followed by sonication. Alternatively, protein lysates were prepared in RIPA buffer containing protease inhibitors and 1% SDS followed by purification with QIAshredder columns (Qiagen). Protein concentration was determined with the Pierce BCA protein assay kit (Thermo Scientific) or the Bio-Rad DC protein assay kit (Bio-Rad, Hercules, CA). Samples were mixed with NuPAGE LDS sample buffer containing NuPAGE sample reducing agent, denatured for 10 min at 70°C and loaded onto NuPAGE Novex 4–12% Bis-Tris protein gels (Invitrogen). Proteins were transferred to PVDF membranes using the Trans-Blot turbo blotting system (Bio-Rad). After blocking for 1 hr in Odyssey blocking buffer, membranes were incubated over night at 4°C with primary antibodies in Odyssey blocking buffer supplemented with 0.1% Tween-20 (Bio-Rad). Secondary antibodies (anti-rabbit IRDye 800CW from LI-COR; anti-mouse Alexa Fluor 680 from Invitrogen) were diluted in Odyssey blocking buffer supplemented with 0.1% Tween-20 and incubated with membranes for 1 hr at room temperature. Proteins were visualized using Odyssey infrared imaging system. Alternatively, membranes were blocked with PBS/0.1% Tween-20/5% milk. Primary and secondary (anti-rabbit HRP or anti-mouse HRP from Thermo Fisher Scientific) antibodies were diluted in PBS/0.1% Tween-20/5% milk and incubated over night at 4°C and for 1 hr at room temperature, respectively. Proteins were visualized using the SuperSignal West Pico chemiluminescent substrate (Thermo Scientific) by autoradiography. Protein quantification was performed using ImageJ and statistical significance was assessed using GraphPad Prism 6. The primary antibodies used in this study are reported in *Supplementary file 1D*.

## Imaging and image analysis

Cells were fixed in 4% paraformaldehyde supplemented with Hoechst 33342. After three washes with PBS, cells were subjected to automated epifluorescence microscopy using an InCell Analyzer 2000 (GE Healthcare, United Kingdom). Four fields per well were imaged using 20 X magnification, DAPI and FITC filters. Images were analyzed with the CellProfiler software (Broad Institute). Briefly, FITC images were segmented using the ilastik tool and a probability map was generated to identify GFP-SQSTM1 dots. Nuclei were identified on the basis of Hoechst 33,342 staining. The total area covered by GFP-SQSTM1 puncta was quantified using the probability map, divided by the nuclei count and averaged across fields. Statistical significance was assed using GraphPad Prism 6.

# Additional information

#### Competing interests

RD, FM, GM, ZW, PB, SL, EF, JA, JSR-H, AL, JKe, CR, JKn, WC, MB, GR, JAT, JAP, CM, LOM, GRH, BN: Employee of Novartis. The other authors declare that no competing interests exist.

## Funding

| Funder | Grant reference number | Author |
| --- | --- | --- |
| Novartis | | Rowena DeJesus<br>Francesca Moretti<br>Gregory McAllister<br>Zuncai Wang<br>Phil Bergman<br>Shanming Liu<br>Elizabeth Frias<br>John Alford<br>John S Reece-Hoyes<br>Alicia Lindeman<br>Jennifer Kelliher<br>Carsten Russ<br>Judith Knehr<br>Walter Carbone<br>Martin Beibel<br>Guglielmo Roma<br>John A Tallarico<br>Jeffery A Porter<br>Craig Mickanin<br>Leon O Murphy<br>Gregory R Hoffman<br>Beat Nyfeler |
| National Institutes of Health | R01DK097485 | Aylwin Ng<br>Ramnik J Xavier |
| National Institutes of Health | P30DK043351 | Aylwin Ng<br>Ramnik J Xavier |
| National Institutes of Health | U19AI109725 | Ramnik J Xavier |

## Author contributions

RD, FM, GM, Conception and design, Acquisition of data, Analysis and interpretation of data, Drafting or revising the article; ZW, PB, SL, Acquisition of data, Analysis and interpretation of data; EF, JA, JSR-H, AL, JKe, CR, JKn, WC, Acquisition of data; MB, AN, RJX, Analysis and interpretation of data; GR, Analysis and interpretation of data, Drafting or revising the article; JAT, JAP, CM, Conception and design; LOM, GRH, BN, Conception and design, Analysis and interpretation of data, Drafting or revising the article

## Author ORCIDs

Walter Carbone, http://orcid.org/0000-0001-6150-8295
Beat Nyfeler, http://orcid.org/0000-0003-0624-9571

# Additional files

### Supplementary files

• Supplementary file 1. Supplementary information on reagents. (A) Schematic representation of the CRISPR vectors used in this study. (B) PCR and sequencing primer sequences for shRNA, genome-wide sgRNA and mini-pool sgRNA libraries. (C) sgRNA sequences selected for validation experiments. (D) Primary antibody information.

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
