## [Decision Letter]

Thank you for submitting your article "Functional CRISPR screening identifies the ufmylation pathway as a regulator of SQSTM1/p62" for consideration by *eLife*. Your article has been favorably evaluated by Arup Chakraborty (Senior editor) and four reviewers, including Narendra Wajapeyee and Michael R Green, who is a member of our Board of Reviewing Editors.

The reviewers have discussed the reviews with one another and the Reviewing Editor has drafted this decision to help you prepare a revised submission

Summary:

DeJesus et al. in this very interesting study perform a genome-wide CRISPR/CAS9-based screening and identified ufmylation pathway as a regulator of SQSTM1/p62. This study also showed that CRISPR-based approach was superior to RNAi at least in this context in identifying the regulators of p62 stability. All of the autophagy assays are appropriate, controlled and consistent with the conclusions of the manuscript. In addition, the experiments and data in support of the conclusion that ufmylation regulates SQSTM and ER stress in an autophagy-independent manner are appropriate. The work is expected to be of general interest to scientists interested in genome-wide screening and specific interest to those in the autophagy field. The sole substantive concern is that the manuscript needs to be written in a more balanced manner as detailed below.

Essential revisions:

1) One concern for the side-by-side comparison is that the standard CRISPR library was not used but rather the authors used a so called "focused CRISPR library", which contains 20 sgRNAs per gene. It is not clear that the standard CRISPR library, which contains 5 sgRNAs per gene, would perform as well and therefore the major conclusion may be misleading. The authors need to revise the text to explicitly discuss this issue.

2) The authors should discuss potential drawbacks of the CRISPR screen such as the inability to obtain positives for genes that are essential for cellular viability. They should also mention that like RNAi, CRISPR/Cas9 can have off-target effects.

3) The authors should discuss that the conclusion that CRISPR screens are superior to RNAi screens cannot be generalized for all scenarios without performing experiments, similar to what the authors have done in this manuscript.

---

## [Author Response]

*Essential revisions:*

1) One concern for the side-by-side comparison is that the standard CRISPR library was not used but rather the authors used a so called "focused CRISPR library", which contains 20 sgRNAs per gene. It is not clear that the standard CRISPR library, which contains 5 sgRNAs per gene, would perform as well and therefore the major conclusion may be misleading. The authors need to revise the text to explicitly discuss this issue.

We have directly compared the performance of the focused library versus the genome-wide library by analyzing the maximal fold change for known modulators of SQSTM1. As shown in Figure 2—figure supplement 1, the genome-wide library with 5 sgRNAs per gene performs equally well in identifying autophagy pathway components as the 20 sgRNAs per gene in the focused library. We have included a sentence with this conclusion in the manuscript (subsection “A genome-wide screen for modulators of SQSTM1 robustly identifies canonical macroautophagy components”, second paragraph).

2) The authors should discuss potential drawbacks of the CRISPR screen such as the inability to obtain positives for genes that are essential for cellular viability. They should also mention that like RNAi, CRISPR/Cas9 can have off-target effects.

We have modified the text to discuss intrinsic limitations of CRISPR screens and updated the text with relevant references (subsection “Impairment of protein ufmylation elicits an ER stress response and induces SQSTM1 expression”, last paragraph).

*3) The authors should discuss that the conclusion that CRISPR screens are superior to RNAi screens cannot be generalized for all scenarios without performing experiments, similar to what the authors have done in this manuscript.*

We have modified the text to discuss that RNAi and CRISPR-based approaches have to be evaluated depending on the screening paradigm and updated the text with relevant references (subsection “Impairment of protein ufmylation elicits an ER stress response and induces SQSTM1 expression”, last paragraph).